# Gle1 is required for tRNA to stimulate Dbp5 ATPase activity in vitro and promote Dbp5-mediated tRNA export in vivo in *Saccharomyces cerevisiae*

**Arvind Arul Nambi Rajan[1], Ryuta Asada[2], Ben Montpetit[1,2]***

[1]Biochemistry, Molecular, Cellular and Developmental Biology Graduate Group, University of California, Davis, Davis, United States; [2]Department of Viticulture and Enology, University of California, Davis, Davis, United States

**Abstract** Cells must maintain a pool of processed and charged transfer RNAs (tRNA) to sustain translation capacity and efficiency. Numerous parallel pathways support the processing and directional movement of tRNA in and out of the nucleus to meet this cellular demand. Recently, several proteins known to control messenger RNA (mRNA) transport were implicated in tRNA export. The DEAD-box Protein 5, Dbp5, is one such example. In this study, genetic and molecular evidence demonstrates that Dbp5 functions parallel to the canonical tRNA export factor Los1. In vivo co-immunoprecipitation data further shows Dbp5 is recruited to tRNA independent of Los1, Msn5 (another tRNA export factor), or Mex67 (mRNA export adaptor), which contrasts with Dbp5 recruitment to mRNA that is abolished upon loss of Mex67 function. However, as with mRNA export, overexpression of Dbp5 dominant-negative mutants indicates a functional ATPase cycle and that binding of Dbp5 to Gle1 is required by Dbp5 to direct tRNA export. Biochemical characterization of the Dbp5 catalytic cycle demonstrates the direct interaction of Dbp5 with tRNA (or double-stranded RNA) does not activate Dbp5 ATPase activity, rather tRNA acts synergistically with Gle1 to fully activate Dbp5. These data suggest a model where Dbp5 directly binds tRNA to mediate export, which is spatially regulated via Dbp5 ATPase activation at nuclear pore complexes by Gle1.

## eLife assessment

The work is a **valuable** contribution to understanding the mechanism of nuclear export of tRNA in budding yeast. The authors present **solid** evidence that Dbp5 functions in parallel with Los1 and Msn5 in tRNA export, in a manner dependent on Gle1 for activation of its ATPase activity but independently of Mex67, Dbp5's partner in mRNA export. It further presents biochemical evidence that Dbp5 can bind tRNA but that Gle1 and InsP6 are required for activating ATP hydrolysis by the Dbp5-tRNA complex, suggesting a possible mechanism for tRNA export by Dbp5.

## Introduction

Key to the production of proteins is the delivery of amino acids to ribosomes by transfer RNAs (tRNAs). Precursor tRNAs (pre-tRNAs) are first transcribed by RNA Polymerase III with a 5′ leader and a 3′ trailer sequence (*Guerrier-Takada et al., 1983*; *Cook et al., 2009*; *Skowronek et al., 2014*; *Harris, 2016*; *Graczyk et al., 2018*). To function in translation, pre-tRNAs must be end matured and modified, spliced, amino-acylated, and exported from the nucleus in eukaryotes. Importantly, it has come to be appreciated that these steps in tRNA processing are not necessarily sequential (*Trotta et al., 1997*;

*For correspondence:
benmontpetit@ucdavis.edu

Competing interest: The authors declare that no competing interests exist.

*Yoshihisa et al., 2003*; *Yoshihisa et al., 2007*; *Wu and Hopper, 2014*; *Hopper and Huang, 2015*; *Wu et al., 2015*; *Phizicky and Hopper, 2023*). For example, in the budding yeast *Saccharomyces cerevisiae*, intron-containing tRNAs are spliced in the cytoplasm on the mitochondrial surface by the tRNA Splicing Endonuclease (SEN) complex following nuclear export (*Yoshihisa et al., 2003*; *Yoshihisa et al., 2007*; *Skowronek et al., 2014*; *Wan and Hopper, 2018*). Furthermore, cytoplasmic tRNAs can be reimported to the nucleus through retrograde transport to allow tRNAs to undergo further modification and quality control which is also used to repress translation during conditions of stress (*Murthi et al., 2010*; *Hopper and Huang, 2015*; *Nostramo and Hopper, 2020*). As tRNAs cycle between the nucleus and cytoplasm, they undergo various chemical modifications that are also spatially regulated (*Phizicky and Hopper, 2023*). The proper processing and modification of tRNAs through this complex regulatory network dictate proper folding of pre-tRNAs and recognition by appropriate tRNA-binding proteins that control subsequent steps in the tRNA life cycle.

In the context of nuclear export, end maturation and amino-acylation are required for recognition by the tRNA Exportin Los1/Exportin-t and Msn5, respectively (*Huang and Hopper, 2015*; *Chatterjee et al., 2017*; *Nostramo and Hopper, 2020*; *Chatterjee et al., 2022*). Although Los1 and Msn5 are the best characterized tRNA export factors, both are non-essential and can be deleted in combination (*Takano et al., 2005*; *Murthi et al., 2010*). Given the essential nature of tRNA export, this has prompted several studies, including a comprehensive screen of nearly all annotated genes in *S. cerevisiae*, to identify additional factors responsible for regulating nucleocytoplasmic dynamics of tRNA localization and processing (*Feng and Hopper, 2002*; *Steiner-Mosonyi et al., 2003*; *Eswara et al., 2009*; *Huang and Hopper, 2015*; *Wu et al., 2015*; *Chatterjee et al., 2017*; *Nostramo and Hopper, 2020*; *Chatterjee et al., 2022*). These studies have elucidated functional roles of the karyopherin Crm1 and mobile nucleoporin Mex67 in regulating tRNA nucleocytoplasmic dynamics (*Chatterjee et al., 2017*; *Derrer et al., 2019*; *Nostramo and Hopper, 2020*; *Chatterjee et al., 2022*). Both factors have previously been shown to function in messenger RNA (mRNA) export (*Hodge et al., 1999*; *Takemura et al., 2004*; *Lund and Guthrie, 2005*; *Derrer et al., 2019*), with Mex67 serving an essential role in the process (*Hodge et al., 1999*; *Lund and Guthrie, 2005*; *Adams and Wente, 2020*). In tRNA export, it is thought that Crm1 and Mex67 both function parallel to Los1 to support export of pre-tRNA and re-export of spliced tRNA that have undergone retrograde transport (*Chatterjee et al., 2017*; *Nostramo and Hopper, 2020*; *Chatterjee et al., 2022*). Moreover, recent reports suggest subspecies specialization and family preferences for tRNA cargo, with Mex67 serving a unique function in the premature export of pre-tRNAs that have not completed 5′ end processing (*Chatterjee et al., 2022*).

While non-coding RNA (ncRNA) and mRNA export pathways have long been hypothesized to be regulated by independent processes, recent studies indicate the participation of numerous mRNA export factors in the regulation of diverse ncRNAs (*Yao et al., 2007*; *Yao et al., 2008*; *Faza et al., 2012*; *Bai et al., 2013*; *Wu et al., 2014*; *Becker et al., 2019*; *Vasianovich et al., 2020*). In addition to tRNA and mRNA, both Mex67 and Crm1 have functions in pre-snRNA, pre-ribosomal subunit, and *TLC1* (telomerase) export (*Yao et al., 2007*; *Faza et al., 2012*; *Bai et al., 2013*; *Wu et al., 2014*; *Becker et al., 2019*; *Vasianovich et al., 2020*). Similarly, in addition to Mex67 and Crm1, the DEAD-box Protein 5 (Dbp5) has also been implicated in each of these RNA export pathways, including tRNA export (*Wu et al., 2014*; *Neumann et al., 2016*; *Mikhailova et al., 2017*; *Becker et al., 2019*; *Lari et al., 2019*). Dbp5 is most well known as an RNA-stimulated ATPase that is spatially regulated by the nucleoporins Nup159 and Gle1 with the small molecule inositol hexakisphosphate (InsP$_6$) at the cytoplasmic face of the nuclear pore complex (NPC) to promote the directional export of mRNA (*Snay-Hodge et al., 1998*; *Tseng et al., 1998*; *Hodge et al., 1999*; *Schmitt et al., 1999*; *Strahm et al., 1999*; *Zhao et al., 2002*; *Weirich et al., 2004*; *Lund and Guthrie, 2005*; *Alcázar-Román et al., 2006*; *Weirich et al., 2006*; *Tran et al., 2007*; *Dossani et al., 2009*; *Fan et al., 2009*; *von Moeller et al., 2009*; *Hodge et al., 2011, Montpetit et al., 2011*; *Noble et al., 2011*; *Kaminski et al., 2013*; *Wong et al., 2016*; *Adams et al., 2017*; *Wong et al., 2018*; *Adams and Wente, 2020*). In addition to these export roles, Dbp5 also shuttles between the nucleus and cytoplasm with reported roles in transcription, R-loop metabolism, and translation (*Hodge et al., 1999*; *Estruch and Cole, 2003*; *Gross et al., 2007*; *Scarcelli et al., 2008*; *Tieg and Krebber, 2013*; *Mikhailova et al., 2017*; *Lari et al., 2019*; *Beißel et al., 2020*). Notably, Nup159 and Gle1 mutants also alter tRNA export (*Lari et al., 2019*), suggesting the mRNA export pathway as a whole (e.g., Mex67, Dbp5, Gle1/InsP$_6$,

Nup159) may support tRNA export. However, a mechanistic understanding of how Dbp5 supports these diverse functions and whether they are regulated by similar co-factors and enzymatic activity is largely not understood.

A recent mutagenesis screen within our research group identified mutants that alter the nucleo-cytoplasmic shuttling dynamics of Dbp5 (*Lari et al., 2019*). Mutations were identified (e.g., *dbp5$^{L12A}$*) that disrupt a nuclear export sequence (NES) recognized by Crm1 to promote Dbp5 transport out of the nucleus. In contrast, another mutant, *dbp5$^{R423A}$*, was found to be impaired in nuclear import from the cytoplasm. Importantly, neither of these mutations disrupt the essential mRNA export functions of Dbp5; however, limited nuclear access of *dbp5$^{R423A}$* induced tRNA export defects, suggesting a potentially novel role for nuclear Dbp5 in tRNA export. In support of this hypothesis, a physical interaction between Dbp5 and tRNA was also characterized, which was elevated upon nuclear accumulation of Dbp5 in a *dbp5$^{L12A}$* mutant (*Lari et al., 2019*).

In this study, genetic and biochemical characterization of Dbp5-mediated tRNA export was performed. The data show that Dbp5 is recruited to pre-tRNA independent of Los1 and Msn5, indicating that Dbp5 has functions independent of the primary Los1-mediated pre-tRNA export pathway. Similarly, Dbp5 does not require Mex67 as an adapter for tRNA binding (unlike proposed mechanisms of mRNA export). In contrast to single-stranded RNA substrates (e.g., mRNA), this study further demonstrates an interaction of Dbp5 with tRNA and double-stranded RNA (dsRNA) that leads to robust ATPase activation only in the presence of Gle1/InsP$_6$. These findings, together with previous research (*Lari et al., 2019*), suggest that Dbp5 engages tRNA within the nucleus in a manner distinct from mRNA and requires Gle1/InsP$_6$ to mediate RNA-stimulated ATPase activation of Dbp5 to promote pre-tRNA export.

## Results
### Dbp5 has functions independent of Los1 in pre-tRNA export

Previous studies have placed the mRNA export factor Mex67, a proposed target of Dbp5 regulation in mRNA export (*Lund and Guthrie, 2005*; *Adams and Wente, 2020*), in a pathway parallel to Los1-regulated pre-tRNA nucleo-cytoplasmic shuttling (*Chatterjee et al., 2017*). To investigate the relationship between Dbp5 and Los1, it was first determined if the subcellular distribution of either protein was dependent on the activity of the other. In a *los1Δ*, the NPC localization of GFP-Dbp5 was not altered (*Figure 1A*). Similarly, when a Dbp5 loss of function was induced using auxin-induced degradation (AID) (*Nishimura et al., 2009*), Los1-GFP localization remained unchanged (*Figure 1B*) while mRNA export defects were detected (*Figure 1—figure supplement 1A*), confirming depletion of Dbp5 to a level sufficient to disrupt mRNA export. To test the genetic relationship of Los1 and Dbp5, a *los1Δ* was combined with the previously identified Dbp5 shuttling mutants (i.e., *dbp5$^{L12A}$* and *dbp5$^{R423}$*). Double mutants of *los1Δ* with *dbp5$^{L12A}$* (increased in the nucleus) or *dbp5$^{R423}$* (depleted from the nucleus) exhibited no growth defects (*Figure 1C*). Similarly, combining *dbp5$^{L12A}$* and *dbp5$^{R423A}$* respectively with *los1Δ/msn5Δ* double mutants (*Figure 1—figure supplement 1B*) revealed minimal growth defects. The viability of these mutants suggests Dbp5 may support Los1 and Msn5 tRNA export pathways or the existence of a parallel pathway(s) that is able to maintain cellular fitness in the presence of *dbp5$^{R423A}$*. Importantly, this genetic relationship between *dbp5$^{R423A}$* and *los1Δ* mirrors those previously published for *mex67-5* and *los1Δ* strain (*Chatterjee et al., 2017*). To further distinguish these possibilities, tRNA fluorescence in situ hybridization (FISH) was performed using a probe specific to the pre-tRNA$^{Ile}_{UAU}$ intron (SRIM03) in single and double mutants. The *los1Δ dbp5$^{R423A}$* double mutant exhibited significantly stronger nuclear accumulation of pre-tRNA compared to single mutants or control after a 4 hr incubation at 37°C (*Figure 1D and E*). Notably, elevating nuclear pools of Dbp5 by combining *dbp5$^{L12A}$* with *los1Δ* did not alter the pre-tRNA export defects observed in *los1Δ* strains, indicating that excess nuclear Dbp5 is not sufficient to suppress the *los1Δ* phenotype. These results were recapitulated with FISH probes targeting precursor/mature isoforms of tRNA$^{Ile}_{UAU}$ and tRNA$^{Tyr}_{GUA}$ (*Figure 1—figure supplement 1C and D*). These data argue for a role for Dbp5 in pre-tRNA export that operates independently of Los1.

To complement the FISH data, northern blotting analyses were conducted to assess tRNA processing. Intron-containing pre-tRNAs are transcribed with 5′ leader and 3′ trailer sequences to generate precursor molecules (P) that are end-processed in the nucleus to generate intron-containing

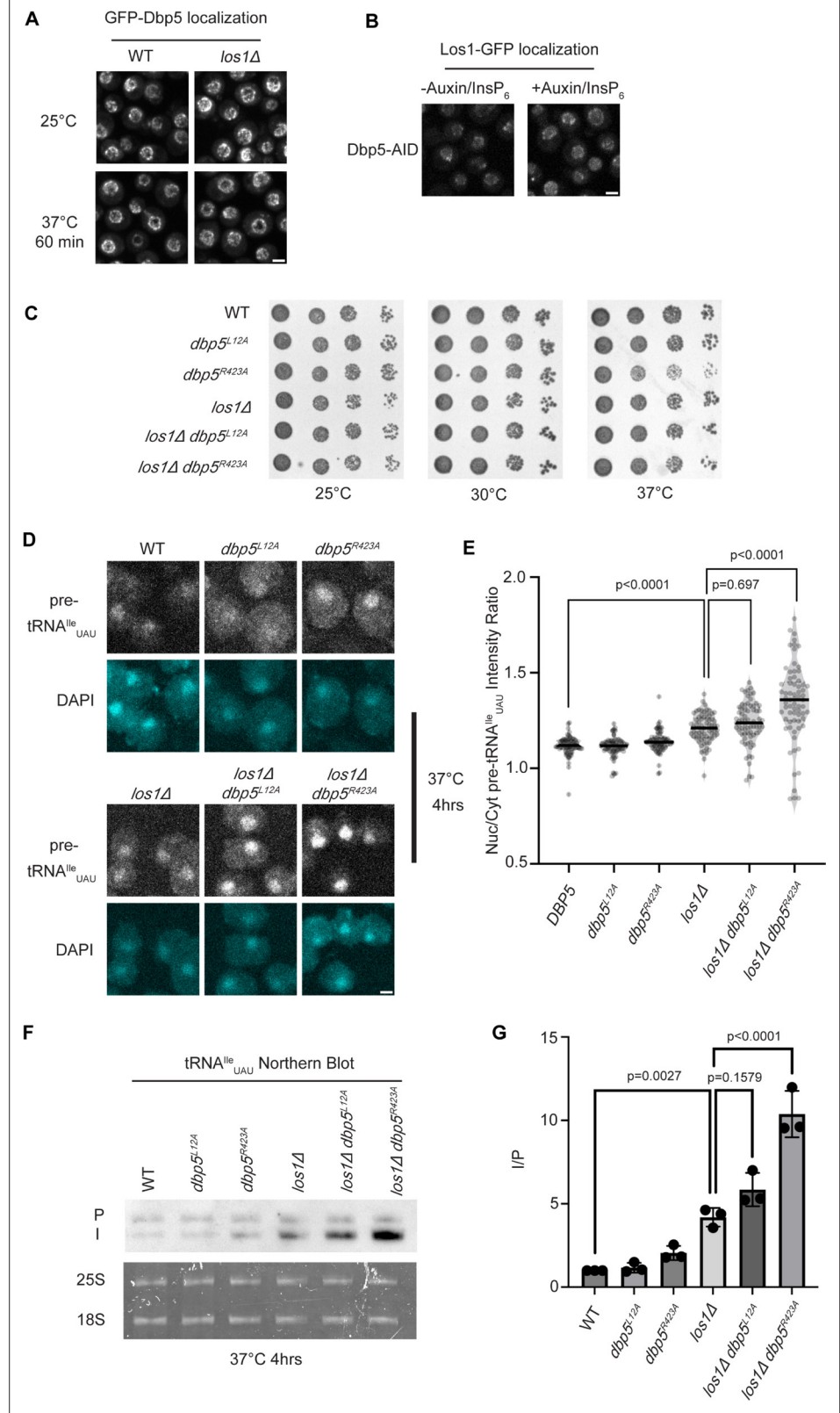

**Figure 1.** Dbp5 functions parallel to Los1 in pre-tRNA export. (**A**) Fluorescent images show GFP-Dbp5 remains enriched at the nuclear periphery in wild-type and *los1Δ* at 25°C and 37°C. Scale bar represents 2 μm. (**B**) Los1-GFP remains nucleoplasmic and associated with nuclear periphery in Dbp5-AID after treatment with DMSO or 500 μM auxin and 10 μM InsP$_6$ for 90 min. Scale bar represents 2 μm. (**C**) Spot assay for growth of strains containing

*Figure 1 continued on next page*

*Figure 1 continued*

untagged *dbp5*[L12A], *dbp5*[R423A], or *los1Δ* integrated at the endogenous gene locus after 2 d at 25, 30, and 37°C on YPD. (**D**) tRNA fluorescence in situ hybridization (FISH) targeting intron of tRNA[Ile]$_{UAU}$ in indicated strains after pre-culture to early log phase at 25°C and shift to 37°C for 4 hr. Scale bar represents 2 µm. (**E**) Quantification of tRNA FISH from (**E**). Ratio of average nuclear to cytoplasmic pixel intensities was calculated across three independent replicate experiments and pooled for plotting. p-Values were calculated using one-way ANOVA. (**F**) Northern blot analysis targeting precursor and mature isoforms of tRNA[Ile]$_{UAU}$. Small RNAs were isolated from strains at mid-log phase growth after pre-culture at 25°C and shift to 37°C for 4 hr. 'P' bands represent intron-containing precursors that have 5' leader/3' trailer sequences, and 'I' bands represent intron-containing end-processed tRNA intermediates that have leader/trailer sequences removed. (**G**) Quantification of northern blot from (**G**). Ratio of signal from intron-containing end-processed intermediates (I) vs 5' leader/3' trailer-containing precursor (P) was calculated and presented relative to I/P ratio observed for WT. Error bars represent standard deviation, and p-values calculated using one-way ANOVA.

The online version of this article includes the following source data and figure supplement(s) for figure 1:

**Source data 1.** Raw data files for northern blot in *Figure 1F*.

**Source data 2.** Uncropped annotated raw northern blot for *Figure 1F*, with relevant bands highlighted.

**Figure supplement 1.** Dbp5 functions parallel to Los1 in pre-tRNA export.

**Figure supplement 1—source data 1.** Raw data files for northern blot in *Figure 1—figure supplement 1E*.

**Figure supplement 1—source data 2.** Uncropped annotated raw northern blot for northern blot in *Figure 1—figure supplement 1E*.

---

intermediates (I) that can be detected by northern blotting (*Wu et al., 2015*). When nuclear export of the end-processed pre-tRNA is disrupted, a change in the precursor (P) to intron-containing intermediate (I) ratio is observed (*Wu et al., 2013*; *Wu et al., 2015*; *Chatterjee et al., 2017*). As such, I/P ratios were measured in the single and *los1Δ/dbp5* double mutants. In line with the increased nuclear localization of pre-tRNAs observed by FISH, an additive defect in pre- tRNA[Ile]$_{UAU}$ and pre-tRNA[Tyr]$_{GUA}$ was observed when *los1Δ* was combined *dbp5*[R423A] (approximately twofold increase in the I/P ratio relative to *los1Δ* alone, *Figure 1F and G*, *Figure 1—figure supplement 1E and F*). These FISH and northern blotting data mirror phenotypes previously reported for mutants of Mex67 in relation to Los1 (*Chatterjee et al., 2022*) and indicate that Dbp5 has functions in pre-tRNA export that are independent of Los1.

## Known tRNA export factors do not recruit Dbp5 to pre-tRNAs in vivo

Dbp5, like other DEAD-box proteins, has been reported to bind nucleic acids through sequence-independent interactions with the phosphate backbone (*Andersen et al., 2006*; *Bono et al., 2006*; *Sengoku et al., 2006*; *Montpetit et al., 2011*). For other helicases of the SF2 family, specificity in RNA substrates is conferred by adaptor proteins (*Lund and Guthrie, 2005*; *Jankowsky, 2011*; *Thoms et al., 2015*). For this reason, it was investigated whether known tRNA export factors aid recruitment of Dbp5 to pre-tRNA substrates. To do so, RNA immunoprecipitation (RIP) experiments were performed with protein-A (prA) tagged Dbp5, integrated at its endogenous locus and present as the sole copy of the gene, in strains where tRNA export factors were deleted. Co-immunoprecipitated RNAs were analyzed by RT-qPCR with primers specific to unspliced intron-containing pre-tRNA[Ile]$_{UAU}$. The abundance of the pre-tRNA[Ile]$_{UAU}$ target in each IP was normalized to the abundance of the target in the corresponding input sample to control for changes in gene expression. Relative enrichment of the target RNA was then compared to the background signal obtained from RNA IPs in a common untagged control. In a *los1Δ* strain, an approximately twofold reduction in the relative amount of pre-tRNA co-immunoprecipitated with Dbp5 (~9.5-fold enrichment in WT to ~4.5-fold in *los1Δ*) was observed, but importantly the deletion failed to abolish the Dbp5 pre-tRNA in vivo interaction (*Figure 2A*). Loss of Msn5, which does not have a published role in the export of intron-containing pre-tRNAs (*Huang and Hopper, 2015*), did not cause a significant change in Dbp5 pre-tRNA interactions by RNA IP. These results indicate that Los1 may support Dbp5-pre-tRNA complex formation, but Dbp5 maintains an ability to bind pre-tRNA in vivo in the absence of Los1. This finding is consistent with the additive tRNA export defects observed in *los1Δ dbp5*[R423A] double mutants (*Figure 1*) and support the hypothesis that Dbp5 has a function in tRNA export parallel to Los1.

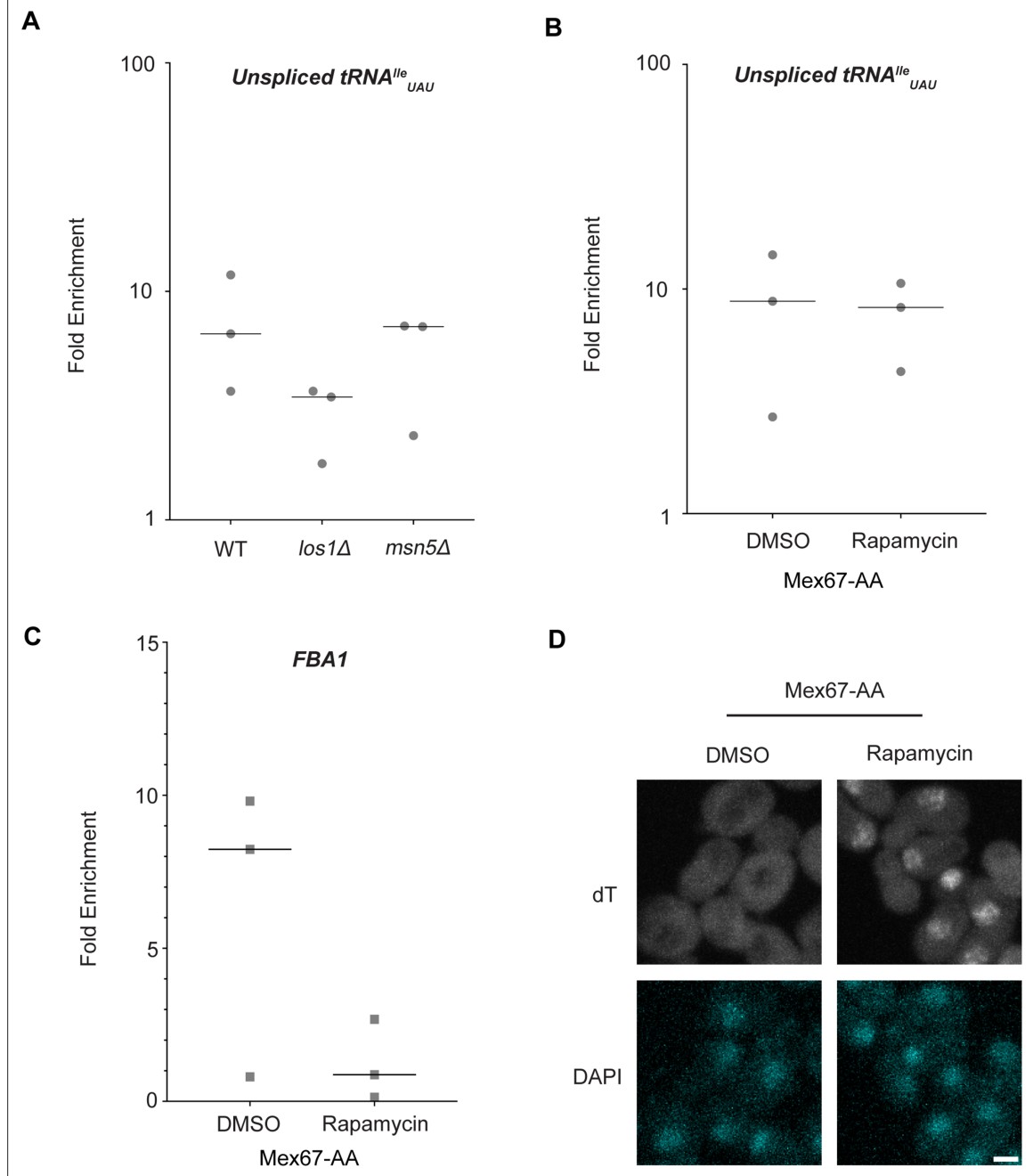

**Figure 2.** Los1 and Mex67 are not required for Dbp5 recruitment to pre-tRNA$^{Ile}_{UAU}$. (**A**) Plots show relative fold enrichment of tRNA$^{Ile}_{UAU}$ following prA-Dbp5 RNA IP in a wild-type (WT), *los1Δ*, and *msn5Δ* strain background. Abundance of target gene is normalized to abundance of the target transcript in input samples and represented as fold enrichment relative to RNA IP from an untagged control. (**B**) prA-Dbp5 RNA IP targeting tRNA$^{Ile}_{UAU}$ in Mex67-AA after either treatment with DMSO or 1 μg/ml rapamycin for 15 min. (**C**) prA-Dbp5 RNA IP targeting FBA1 mRNA in Mex67-AA after either treatment with DMSO or 1 μg/ml rapamycin for 15 min. (**D**) dT fluorescence in situ hybridization (FISH) confirming a mRNA export defect caused by loss of function of Mex67 following 15 min incubation with 1 μg/ml rapamycin compared to a DMSO control. Scale bar represents 2 μm.

In mRNA export, Mex67 is a proposed target of Dbp5 activity to promote directional nuclear export (*Adams and Wente, 2020*). Since neither of the non-essential tRNA export factors abolished Dbp5-tRNA interaction when deleted, it was tested if Mex67 has a role in recruiting Dbp5 to tRNA. An Anchors Away approach was used to rapidly re-localize Mex67 to the cytoplasmic peroxisomal anchor (Pex25-FKBP12) in an inducible manner dependent on the addition of rapamycin (*Haruki et al., 2008*). The resulting strains were sensitive to rapamycin, causing nuclear mRNA accumulation by dT

FISH within 15 min of addition (*Figure 2D*). This timing is consistent with a previous report employing Mex67 Anchors Away (Mex67-AA) (*Tudek et al., 2018*). PrA-Dbp5 was integrated at its endogenous locus in these strains, and RNA-IP experiments were performed before and after rapamycin addition as described above. Under these conditions of Mex67-AA re-localization, no significant change in the Dbp5-tRNA$^{Ile}_{UAU}$ interaction was observed (median of ~8.8-fold enrichment before addition of rapamycin and ~8.2-fold after, *Figure 2B*). Importantly, consistent with the function of Mex67 in mRNA export (*Snay-Hodge et al., 1998*; *Tseng et al., 1998*; *Schmitt et al., 1999*; *Strahm et al., 1999*; *Weirich et al., 2004*; *Lund and Guthrie, 2005*; *Tran et al., 2007*; *Dossani et al., 2009*; *von Moeller et al., 2009*; *Adams and Wente, 2020*), the nuclear retention of mRNAs via Mex67-AA resulted in strongly reduced binding of Dbp5 to the *FBA1* mRNA (from a median of approximately eight-fold enrichment above background before addition of rapamycin to no enrichment after, *Figure 2C*). Together, these RNA IP experiments suggest that Dbp5 binds to tRNAs independent of known tRNA export proteins and does so in a manner that is distinct from mRNA.

## The Dbp5 ATPase cycle supports tRNA export in vivo

In mRNA export, RNA binding, Gle1 activation, and ATP hydrolysis by Dbp5 are all critical for directional transport (*Hodge et al., 2011*, *Noble et al., 2011*). The *dbp5$^{R426Q}$* and *dbp5$^{R369G}$* mutants have previously been shown to exhibit deficient RNA binding (<5% of WT), ATPase activity (~10 and 60% of WT, respectively) and have a dominant-negative effects on mRNA export status when overexpressed (*Hodge et al., 2011*, *Noble et al., 2011*). Additionally, it has been shown that *dbp5$^{R369G}$* has elevated Gle1 affinity and effectively competes with WT Dbp5 for Gle1 binding (*Hodge et al., 2011*). In contrast, *dbp5$^{E240Q}$* is an ATP hydrolysis mutant that is competent to bind RNA as it is biased to the ATP bound state, but is recessive lethal showing no mRNA export defect when overexpressed in the context of endogenously expressed WT Dbp5 (*Hodge et al., 2011*). The *dbp5$^{R426Q}$* and *dbp5$^{R369G}$* mutants localize to the nuclear periphery similar to wild-type Dbp5; however, *dbp5$^{E240Q}$* is predominantly cytoplasmic (with some nuclear envelope localization detectable when wild-type Dbp5 is depleted) (*Hodge et al., 2011*). Unlike the impact of these mutants on mRNA export, it has been reported that the ability of Dbp5 to complete ATP hydrolysis (but not RNA binding) is dispensable for pre-ribosomal subunit export, as is Gle1 (*Neumann et al., 2016*).

Based on the observation that Gle1 functions to support tRNA export (*Lari et al., 2019*), these Dbp5 mutants deficient in ATPase activity or altered Gle1 binding were tested for their influence on tRNA export to infer if tRNA export is more similar to mRNA or pre-ribosomal subunit export. To individually express these lethal mutants and the wild-type control, *DBP5*, *dbp5$^{R426Q}$*, *dbp5$^{R369G}$*, or *dbp5$^{E240Q}$* were integrated at the *URA3* locus under regulation of the inducible pGAL promoter. Protein expression was induced for 6 hr by shifting cells from raffinose to galactose containing media. Induction was followed by 1 hr of growth in glucose to halt expression and relieve potential changes in tRNA export induced by a change in the primary carbon source. Using this approach, all proteins were expressed as indicated by western blotting (*Figure 3A*) and showed the previously reported impact on mRNA export (*Figure 3B*). Northern blotting targeting tRNA$^{Ile}_{UAU}$ revealed both *dbp5$^{R426Q}$* and *dbp5$^{R369G}$*, but not *dbp5$^{E240Q}$*, showed an accumulation of intron-containing precursor tRNA$^{Ile}_{UAU}$ compared to overexpression of wild-type Dbp5 (I/P ratio of 1.97 ± 0.06 vs 1.93 ± 0.31 vs 1.32 ± 0.41, respectively) (*Figure 3C and D*). These phenotypes differ from the reported impacts on pre-ribosomal subunit transport, indicating Dbp5 functions differently in these non-coding RNA export pathways. Furthermore, these data indicate that both the ATPase cycle and regulation of Dbp5 by Gle1 are central to the function of Dbp5 in tRNA export, as each activity is in mRNA export.

## Dbp5 ATPase activity in the presence of tRNA is Gle1-dependent

Given the impact of *dbp5$^{R426Q}$* and *dbp5$^{R369G}$* overexpression on tRNA export in vivo, the Dbp5 ATPase cycle and role of Gle1 in the context of tRNA were further investigated in vitro. Previous studies have characterized Dbp5 binding to single-stranded RNA (ssRNA) substrates and demonstrated Dbp5 binding to ssRNA is linked to the nucleotide state of the enzyme, with the highest affinity for ssRNA occurring when Dbp5 is ATP bound (*Weirich et al., 2006*; *Montpetit et al., 2011*; *Arul Nambi Rajan and Montpetit, 2021*). Following ssRNA binding, hydrolysis of ATP to ADP results in a reduction in the affinity of Dbp5 for the ssRNA substrate (*Weirich et al., 2006*). While there has been extensive characterization of the structural and biochemical details of Dbp5, ssRNA, and co-regulators (*Wong*

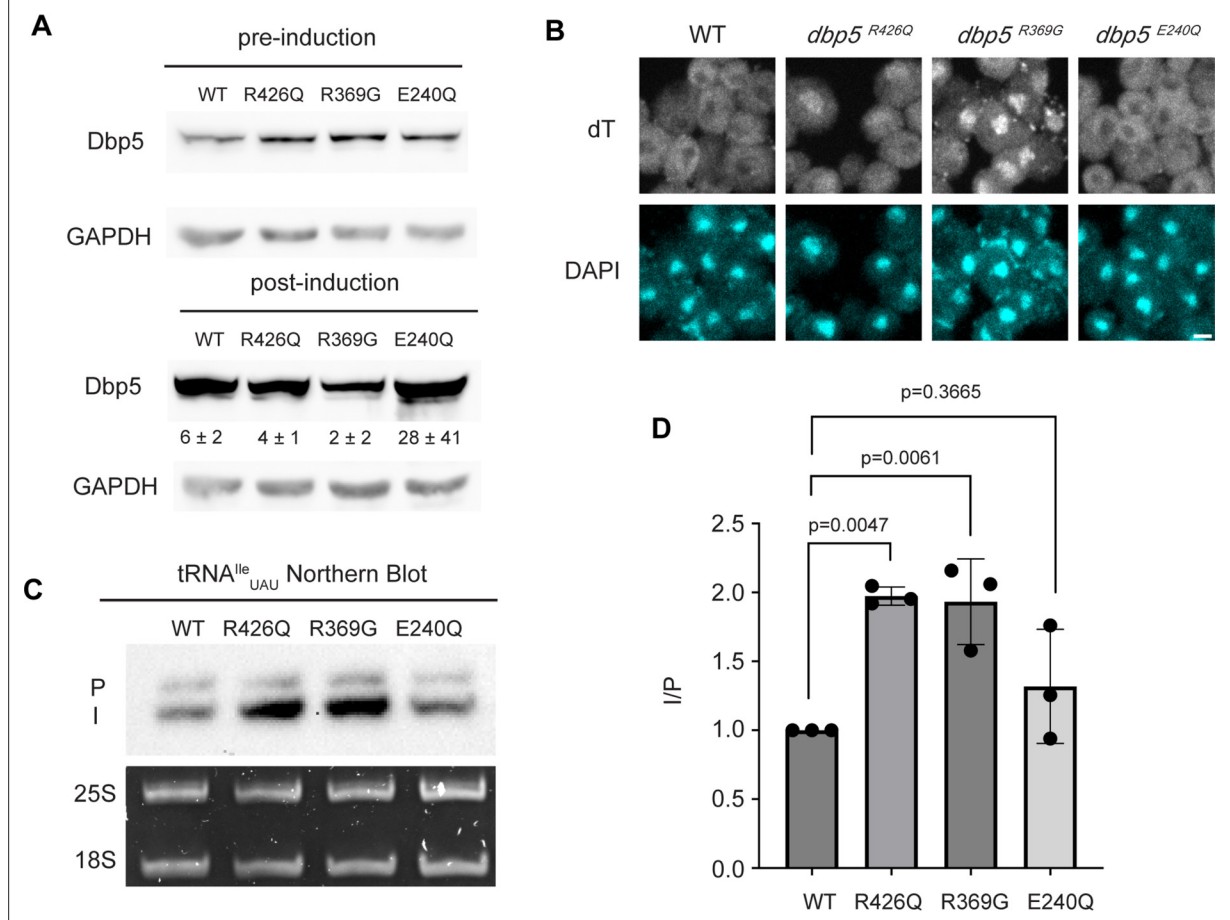

**Figure 3.** ATPase activity and Gle1 are required for Dbp5-mediated tRNA export in vivo. (**A**) Western blot with either mouse monoclonal anti-DBP5 or anti-GAPDH (loading control) antibody to detect overexpression of untagged Dbp5 and Dbp5 ATPase mutants. Strains were cultured in raffinose at 25°C to derepress pGAL promoter (pre-induction), and expression was induced by shifting cultures into 2% galactose containing media for 6 hr (post-induction). Expression was stopped by addition of glucose for 1 hr. Overexpression was observed over the level of wild-type Dbp5 that is expressed from the endogenous locus and is present in the pre-induction samples. Top and bottom panels are from the same blot and processed equally. Quantification of the extent overexpression in post-induction samples is presented below Dbp5 bands. Signal intensity of Dbp5 bands was normalized to abundance of Gapdh and presented relative to pre-induction levels of expression. (**B**) dT fluorescence in situ hybridization (FISH) confirms previously reported mRNA export status phenotypes for Dbp5 ATPase mutants, with *dbp5*$^{R426Q}$ and *dbp5*$^{R369G}$ showing a nuclear accumulation of poly(A)-RNA. Scale bar represents 2 μm. (**C**) Northern blot analysis targeting precursor and mature isoforms of tRNA$^{Ile}_{UAU}$ from yeast strains overexpressing *DBP5*, *dbp5*$^{R426Q}$, *dbp5*$^{R369G}$, or *dbp5*$^{E240Q}$. (**D**) Quantification of northern blot from (**C**). Ratio of signal from intron-containing end-processed intermediates (I) vs 5' leader/3' trailer-containing precursor (P) was calculated and presented relative to I/P ratio observed for WT. Error bars represent standard deviation, and p-values calculated using one-way ANOVA.

The online version of this article includes the following source data for figure 3:

**Source data 1.** Raw data files for western and northern blots in *Figure 3A and C*.

**Source data 2.** Uncropped annotated raw western and northern blots for *Figure 3A and C*, with relevant bands highlighted.

et al., 2018; *Mason and Wente, 2020*; *Gray et al., 2022*), no such understanding exists for Dbp5 engaging highly structured RNA substrates like tRNA. As such, it was first tested if recombinant full-length Dbp5 could bind commercially available yeast mixed tRNAs or the yeast phenylalanine tRNA (both substrates that have been used in previous biochemical and structural studies; *Shi and Moore, 2000*; *Yao et al., 2007*).

To test whether Dbp5 can form complexes with tRNA in vitro, electrophoretic mobility shift assays (EMSA) were performed in the presence of different nucleotide states and tRNA substrates. Consistent with published observations of complex formation between Dbp5 and ssRNA (*Weirich et al., 2006*; *Montpetit et al., 2011*), Dbp5 generated band shifts for both mixed tRNA substrates and Phe tRNA in the presence of the ATP state analog ADP•BeF$_3$ (*Figure 4A*). In the presence of ATP or

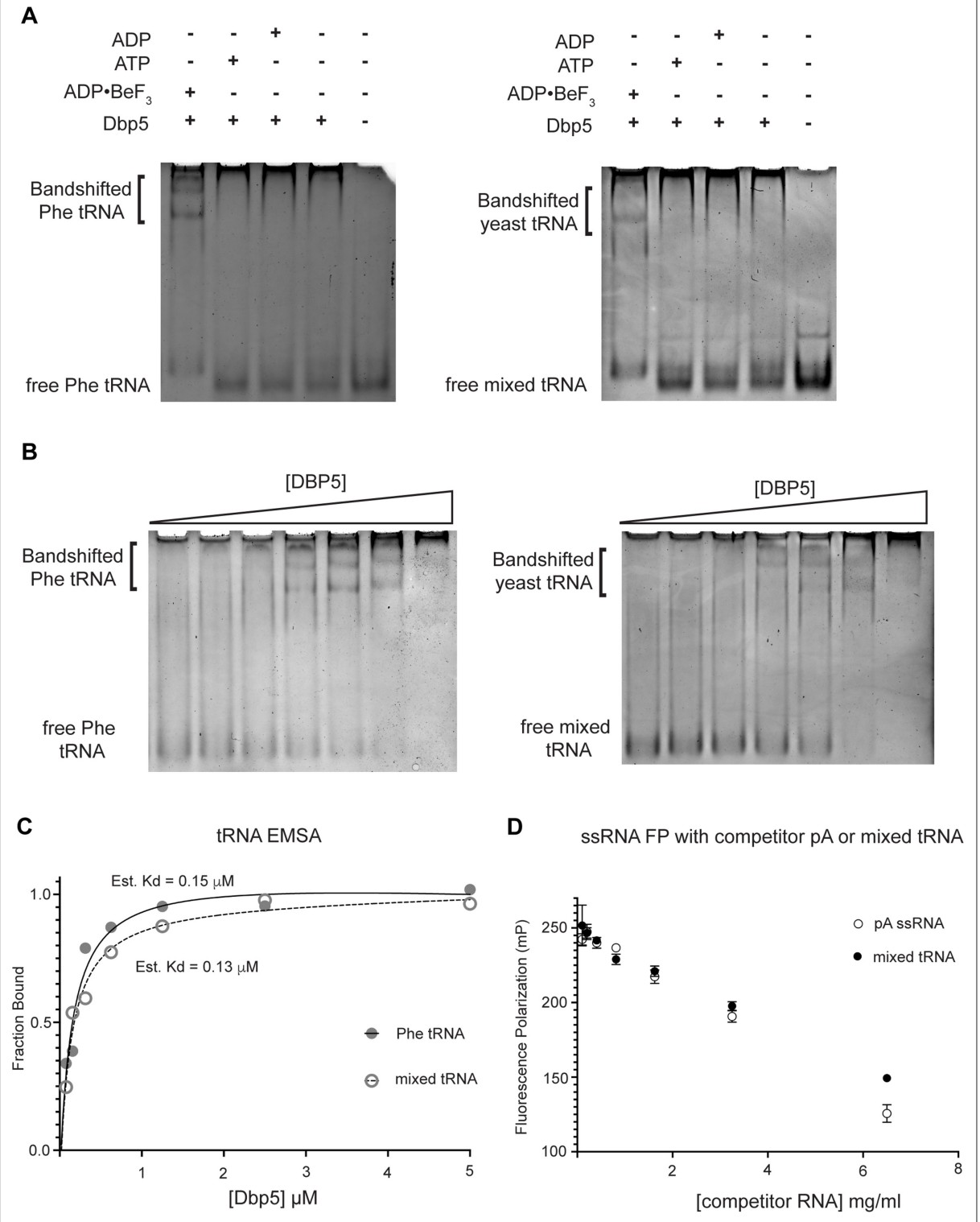

**Figure 4.** Dbp5 binds yeast tRNA in vitro. (**A**) Full-length recombinant Dbp5 (fl-Dbp5) binds both mixed yeast tRNA and phenylalanine (Phe) tRNA in the presence of the ATP mimetic ADP•BeF$_3$ by electrophoresis mobility shift assay (EMSA). RNA-binding reaction was conducted in the presence of ADP•BeF$_3$, ATP, ADP, or no nucleotide and resolved on 6% native polyacrylamide gel at 70 V. Reactions contained 2 μM fl-Dbp5, 1 mM nucleotide when present, 250 ng tRNA, and binding buffer. (**B**) EMSA experiments in which increasing concentrations of fl-Dbp5 (1:2 dilution series starting at 5 uM Dbp5) were titrated into RNA-binding reactions in the presence of 1 mM ADP•BeF$_3$, 250 ng mixed yeast tRNA or Phe tRNA, and binding buffer. (**C**) Band intensities of free probe and band shifts were quantified, and bound fraction was calculated for each well of EMSAs in (**B**). One site binding model was

*Figure 4 continued on next page*

*Figure 4 continued*

fit to the data using GraphPad Prism to estimate Kd for both mixed yeast tRNA and Phe tRNA. (D) Unlabeled pA ssRNA and mixed yeast tRNA compete with a 16nt fluorescein-labeled ssRNA for fl-Dbp5 binding. Fluorescence polarization competition assays were performed by titrating increasing concentration of an unlabeled competitor, pA (open square) or mixed yeast tRNA (closed circle), in reactions containing 50 nM fluorescein-labeled ssRNA, 2.5 mM ADP•BeF$_3$, 1 µM Dbp5, and buffer. Error bars represent standard deviation of three independent experiments.

The online version of this article includes the following source data for figure 4:

**Source data 1.** Raw data files for electrophoretic mobility shift assays (EMSAs) in *Figure 4A and B* .

**Source data 2.** Uncropped annotated electrophoretic mobility shift assays (EMSAs) for *Figure 4A and B*, with relevant bands highlighted.

ADP, or with no nucleotide, Dbp5 failed to form a similar band shift, which exactly parallels reported interactions between Dbp5 and ssRNA in these nucleotide states (*Weirich et al., 2006*; *Montpetit et al., 2011*).

To further estimate an affinity for tRNA, both the phenylalanine tRNA and yeast mixed tRNAs were used in EMSA experiments with varying concentrations of Dbp5 (and fixed concentrations of ADP•BeF$_3$ and tRNA). From quantification of EMSAs, an estimated K$_d$ of ~150 nM (Phe tRNA) and ~130 nM (mixed tRNA) was obtained (*Figure 4B and C*). In addition to EMSAs, Dbp5 binding to tRNA was observed and tested in competition assays using fluorescence polarization measurements. In these assays, increasing concentrations of unlabeled tRNA or an ssRNA homopolymeric polyadenylic acid (pA, routinely used for RNA activation in steady-state ATPase assays; *Weirich et al., 2006*) were found to compete with a fluorescently labeled ssRNA for Dbp5 in the presence of ADP•BeF$_3$ (*Figure 4D*). These data, from orthogonal approaches, indicate that Dbp5 interacts directly with tRNA in vitro and does so in a manner governed by principles that resemble Dbp5-ssRNA binding.

Interaction with ssRNA substrates has been well documented to stimulate the Dbp5 ATPase cycle (*Weirich et al., 2004*; *Lund and Guthrie, 2005*; *Alcázar-Román et al., 2006*; *Weirich et al., 2006*; *Tran et al., 2007*; *Dossani et al., 2009*; *von Moeller et al., 2009*; *Hodge et al., 2011*, *Montpetit et al., 2011*; *Noble et al., 2011*). As such, it was next determined if binding of tRNA to Dbp5 stimulates ATPase activity using a spectrophotometric ATPase assay (*Weirich et al., 2006*; *Montpetit et al., 2011*). Unlike an ssRNA substrate (i.e., pA) that can maximally stimulate ATP turnover to ~0.5 ATP/s, neither tRNA nor poly(I:C) (used as an orthogonal dsRNA substrate) stimulated Dbp5 ATPase activity over a range of RNA concentrations (*Figure 5A*). These results with the above binding data suggest Dbp5 can engage structured and dsRNA substrates, but this does not lead to productive ATP hydrolysis.

The nucleoporin Gle1 with the small molecule inositol hexakisphosphate (InsP$_6$) has been shown to synergistically activate Dbp5 ATPase activity at low RNA concentrations (*Weirich et al., 2006*; *Montpetit et al., 2011*). These findings have led to a model of spatially regulated Dbp5 activity to promote mRNA-protein complex remodeling at the cytoplasmic face of an NPC where Gle1 is localized, resulting in directional mRNA export (*Hodge et al., 1999*; *Strahm et al., 1999*; *Alcázar-Román et al., 2006*; *Weirich et al., 2006*; *Dossani et al., 2009*; *Hodge et al., 2011*, *Montpetit et al., 2011*; *Noble et al., 2011*; *Adams et al., 2017*; *Wong et al., 2018*; *Arul Nambi Rajan and Montpetit, 2021*). In the context of tRNA, addition of Gle1/InsP$_6$ maximally stimulated Dbp5 (1.03 ± 0.04 ATP/s) to a level like that of ssRNA (1.11 ± 0.07) (*Figure 5B*). The synergistic activation of Dbp5 ATPase activity by Gle1/InsP$_6$ and tRNA is mirrored by poly(I:C) and observed with both mixed yeast tRNAs as well as yeast phenylalanine tRNA (*Figure 5B*). To confirm that this enhanced RNA activation by Gle1/InsP$_6$ was not the result of contaminating ssRNA, ATPase assays were performed after treatment of tRNA or poly (I:C) with RNase T1 for 2 hr at 37°C. RNase T1 degrades ssRNA, not dsRNA, allowing determination of whether RNA activation observed in the presence of Gle1/InsP$_6$ is specific to properly folded tRNA and poly(I:C). As expected, treatment of the ssRNA (pA) with RNase T1 resulted in the loss of RNA-stimulated Dbp5 ATPase activity (0.68 ± 0.06 before RNase T1 treatment to 0.17 ± 0.01 after). Furthermore, RNase T1 treatment reduced pA/Gle1/InsP$_6$ activation (0.98 ± 0.17 before treatment to 0.69 ± 0.07 after) to that of Dbp5 and Gle1/InsP$_6$ alone (0.62 ± 0.03) (*Figure 5C*). However, RNase T1-treated tRNA remained unchanged in the ability to activate Dbp5 in the presence of Gle1/InsP$_6$ (1.19 + 0.11 without vs 1.18 ± 0.19 ATP/s with RNase T1 treatment). This was recapitulated with poly(I:C) (1.26 ± 0.11 ATP/s without and 1.46 ± 0.135 ATP/s with RNase T1), confirming that the synergistic activation of Dbp5 observed in the presence of Gle1/InsP$_6$ is dependent on the structured tRNA or dsRNA. These findings provide for the possibility that Dbp5 engages tRNA in the

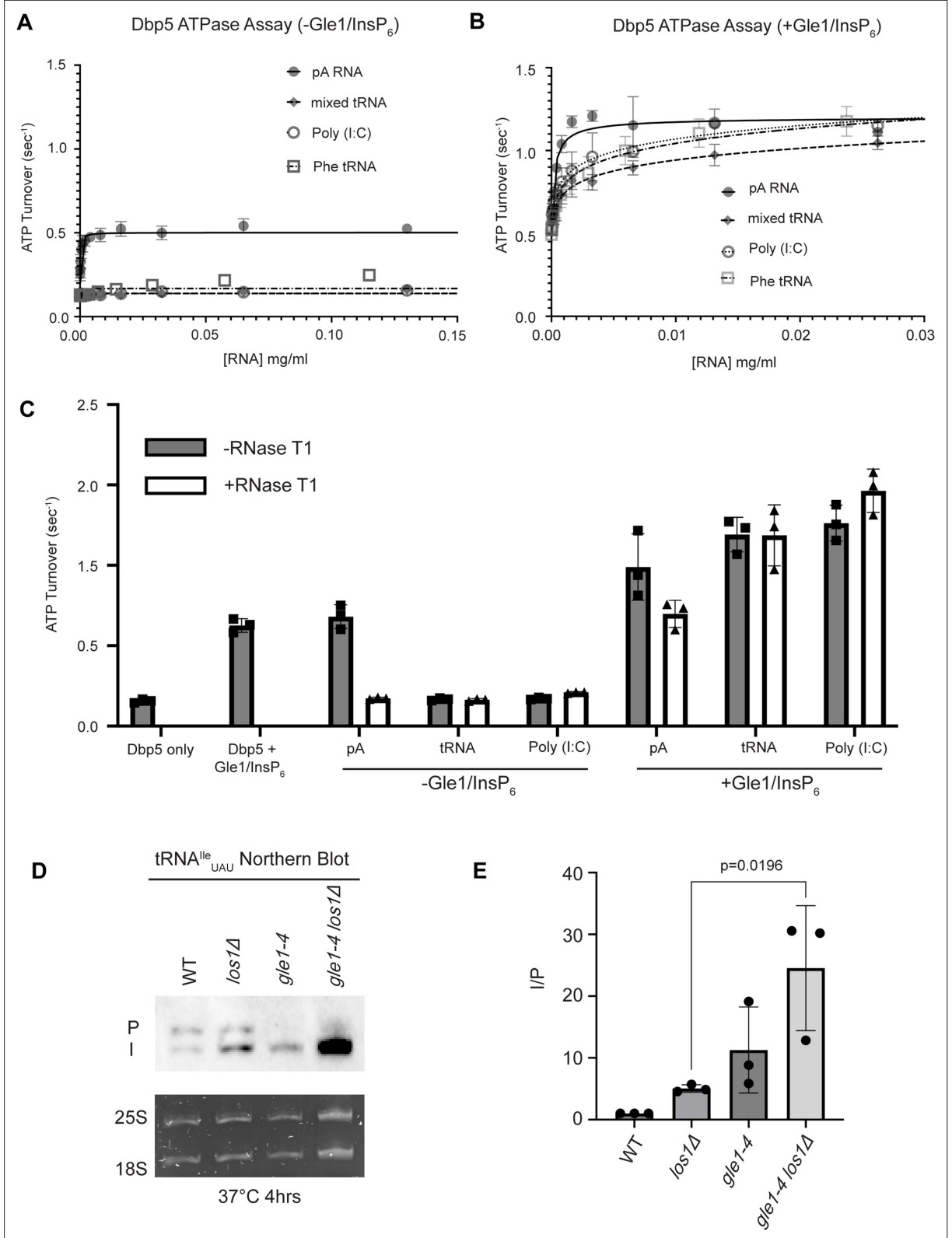

**Figure 5.** tRNA alone does not stimulate the Dbp5 ATPase cycle but can act synergistically with Gle1/InsP$_6$ to fully activate Dbp5. (**A**) ATPase activity of Dbp5 is stimulated by ssRNA (pA, closed square) but not mixed yeast tRNA (closed diamond), Phe tRNA (open square), or poly (I:C) dsRNA (open circle). Steady-state ATPase assays were conducted in the presence of 1 µM Dbp5, 2.5 mM ATP, and varying concentrations of RNA. Data was fit to Michaelis–Menten equation using GraphPad Prism. Error bars represent standard deviation of three independent experiments. (**B**) Gle1/InsP$_6$ synergistically

*Figure 5 continued on next page*

*Figure 5 continued*

stimulates Dbp5 ATPase activity with mixed yeast tRNA (closed diamond), Phe tRNA (open square), and poly (I:C) dsRNA (open circle) substrates like ssRNA (pA). Steady-state ATPase assays were conducted in the presence of 1 µM Dbp5, 2.5 mM ATP, 2 µM Gle1, 2 µM InsP$_6$, and varying concentration of RNA. Data was fit to an allosteric sigmoidal model in GraphPad Prism. Error bars represent standard deviation of three independent experiments. (**C**) RNase T1 treatment of RNA for 2 hr at 37°C prior to ATPase assays confirms that the observed synergistic activation of Dbp5 ATPase activity by Gle1/InsP$_6$ and tRNA or dsRNA is not caused by low levels of contaminating ssRNA. Steady-state ATPase assays were conducted in the presence of 1 µM Dbp5, 2.5 mM ATP, 0.2 mg/ml RNA, and 2 µM Gle1/InsP$_6$ when indicated. (**D**) Northern blot analysis targeting precursor and mature isoforms of tRNA$^{Ile}_{UAU}$. Small RNAs were isolated from strains at mid-log phase growth after pre-culture at 25°C and shift to 37°C for 4 hr. 'P' bands represent intron-containing precursors that have 5′ leader/3′ trailer sequences, and 'I' bands represent intron-containing end-processed tRNA intermediates that have leader/trailer sequences removed. (**E**) Quantification of northern blot from (**D**). Ratio of signal from intron-containing end-processed intermediates (I) vs 5′ leader/3′ trailer-containing precursor (P) was calculated and presented relative to I/P ratio observed for WT. Error bars represent standard deviation, and p-values calculated using one-way ANOVA.

The online version of this article includes the following source data for figure 5:

**Source data 1.** Raw data files for northern blot in *Figure 5D*.

**Source data 2.** Uncropped annotated raw northern blot from *Figure 5D*, with relevant bands highlighted.

cell without ATPase activation and could remain bound until the Dbp5-tRNA complex reaches Gle1 at the cytoplasmic side of an NPC. In support of this model, a strong additive defect in pre- tRNA$^{Ile}_{UAU}$ processing was observed by northern blotting when *los1Δ* was combined *gle1-4* (approximately five-fold increase in the I/P ratio relative to *los1Δ* alone, *Figure 5D and E*) after a 4 hr incubation at 37°C. These data indicate that Gle1, like Dbp5, supports pre-tRNA export independent of Los1.

## Discussion

The best characterized tRNA export factors in *S. cerevisiae*, Los1 and Msn5, are non-essential and the *los1Δ/msn5Δ* double mutant is viable despite the essential role of tRNA export (*Takano et al., 2005*; *Murthi et al., 2010*). These data suggest additional mechanisms for regulated tRNA export, which is supported by recent publications that have implicated mRNA and rRNA export factors such as Mex67, Nup159, Gle1, Dbp5, and Crm1 in tRNA export (*Wu et al., 2015*; *Chatterjee et al., 2017*; *Nostramo and Hopper, 2020*; *Chatterjee et al., 2022*). However, a mechanistic understanding of how these factors function in tRNA export and how such roles differ or relate to previously characterized roles in RNA export is still unresolved. Here, a combination of in vivo and in vitro data provides evidence that (1) Dbp5 and Gle1 have functions independent of Los1 in tRNA export; (2) Dbp5 can bind tRNA directly in vitro and does so in a manner that does not require Los1, Msn5, and Mex67 in vivo; and (3) the Dbp5 ATPase cycle is uniquely modulated by Gle1/InsP$_6$ in the presence of structured RNA (e.g., tRNA or dsRNA) in vitro and the ATPase cycle supports tRNA export in vivo.

DEAD-box proteins (including Dbp5) broadly interact with RNA via the phosphate backbone (*Andersen et al., 2006*; *Andersen et al., 2006*; *Linder, 2006*; *Sengoku et al., 2006*; *Linder and Fuller-Pace, 2013*, *Arul Nambi Rajan and Montpetit, 2021*). The observed Dbp5-tRNA interaction is nucleotide dependent, supporting a specificity to the observed binding (*Figure 4A and B*). Further-more, structural data has elucidated the synergistic activation of Dbp5 ATPase cycle induced by Gle1/InsP$_6$ and mRNA binding (*Montpetit et al., 2011*). While further structural analysis is critical for under-standing the same synergistic ATPase activation observed with tRNA and Gle1/InsP$_6$, these results also support direct and productive binding in vitro with a structured RNA. The synergistic activation of Dbp5 by poly (I:C) in the presence of Gle1/InsP$_6$ suggests the interaction is sequence independent and is not driven by elements unique to tRNAs. Collectively, the in vitro characterization of Dbp5 and tRNA described here suggests two non-mutually exclusive possibilities for why Dbp5 is only activated by tRNA/dsRNA in the presence of Gle1/InsP$_6$. First, it may be that Dbp5 binds tRNA to form an inhibited intermediate that is relieved by Gle1/InsP$_6$. This is an attractive hypothesis that is unique from how Dbp5 engages ssRNA (e.g., mRNA) and potentially supports previously proposed export models that suggest Dbp5 may bind and travel with a tRNA from nucleus to cytoplasm (*Lari et al., 2019*). A second possibility is that Gle1/InsP$_6$ enhances the affinity of Dbp5 for tRNA and 'non-mRNA' substrates, which is supported by biochemical characterizations that show Gle1/InsP$_6$ promotes a shift in the steady-state distribution of populated Dbp5 intermediates from weak to strong RNA-binding states (*Hodge et al., 2011*, *Montpetit et al., 2011*; *Noble et al., 2011*; *Wong et al., 2016*; *Wong*

*et al., 2018*; *Gray et al., 2022*). Future structural and biochemical studies are expected to distinguish among these possibilities. Moreover, Dbp5 was previously reported to not bind dsRNA substrates (*Weirich et al., 2006*); as such the findings that Dbp5 can bind, and in the presence of Gle1/InsP$_6$, be activated by tRNA should motivate reevaluation of possible functions of Dbp5-Gle1/InsP$_6$ in regulating other highly structured RNA or 'non-canonical' substrates in vivo.

While it has been reported previously that tRNA fails to stimulate Dbp5 ATPase cycle (*Tseng et al., 1998*), here it is shown that despite a lack of stimulation when Gle1/InsP$_6$ is absent, Dbp5 is able to bind tRNA in vitro and in vivo. The ability of a tRNA to bind but not stimulate the ATPase cycle of Dbp5 has important implications for mechanisms by which Dbp5 may function in tRNA export. For example, in mRNA export it is thought that Dbp5 functions are localized at NPCs (*Derrer et al., 2019*; *Lari et al., 2019*; *Adams and Wente, 2020*), where the Dbp5 ATPase cycle is tightly regulated by co-factors Nup159 and Gle1/InsP$_6$ (*Hodge et al., 1999*; *Rollenhagen et al., 2004*; *Hodge et al., 2011*, *Montpetit et al., 2011*; *Noble et al., 2011*; *Wong et al., 2018*). In fact, recent studies have shown the essential function of both Dbp5 and Mex67 in mRNA export can be accomplished when these proteins are fused to nuclear pore components; in addition, Dbp5 does not form a complex

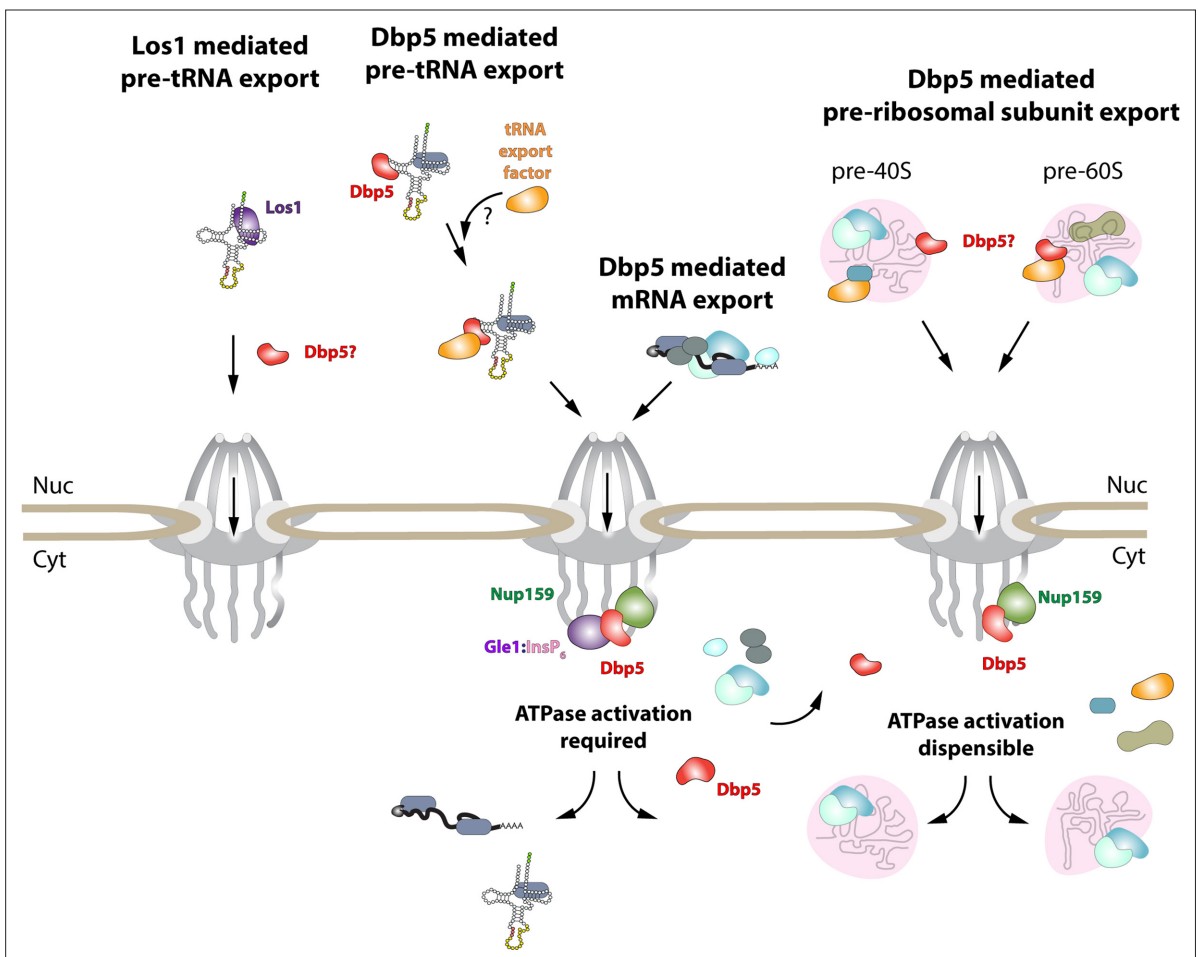

**Figure 6.** Model of Dbp5 function in mRNA, tRNA, and pre-ribosomal subunit export. Data from this study support a model in which Dbp5-mediated export of both mRNA and tRNA require Gle1/InsP$_6$ activation to remodel RNPs at the cytoplasmic face of the nuclear pore complex (NPC). For tRNA export, lack of RNA-mediated ATPase stimulation following RNA binding may lead to the formation of tRNA-bound intermediates that act as adapters for recruitment of yet to be identified transport factors. For mRNA export, it has been shown that Dbp5 function is limited to the nuclear periphery and Dbp5 does not form stable nuclear complexes with mRNA. Gle1/InsP$_6$-mediated activation of Dbp5 catalytic cycle and RNA release likely promotes recycling of export factors and Dbp5 to function in further rounds of export or other functions. In contrast, for Dbp5-mediated pre-ribosomal subunit export Dbp5 ATPase cycle and Gle1/InsP$_6$ stimulation are dispensable for transport. As such, RNA binding may promote export through an unknown mechanism and that Dbp5-mediated remodeling does not occur at NPCs, with factors such as Mex67 persisting on pre-ribosomal subunits following export to the cytoplasm. While a role for Dbp5 functioning in the Los1-mediated pre-tRNA export pathway cannot be excluded, the data presented in this study support a Dbp5-mediated tRNA export pathway that exists independent of and parallel to Los1.

with mRNAs in the nucleus (*Derrer et al., 2019*; *Adams and Wente, 2020*). These reports suggest that Dbp5 does not travel with an mRNA from the nucleoplasm to the cytoplasmic face of an NPC as part of an mRNA-protein complex; rather, Dbp5 likely engages mRNAs after they exit the NPC transport channel to direct the final step(s) of mRNA export (*Figure 6*). However, in contrast to this mRNA export model, the unique ability of Dbp5 to bind tRNA without ATP stimulation, and then be fully activated by Gle1/InsP$_6$, presents the potential for a novel mechanism of function in regulating tRNA export. Namely, that Dbp5 may enter the nucleus to stably engage tRNA and direct events leading to export, following which Dbp5 is removed from the tRNA by Gle1/InsP$_6$ upon nuclear exit (*Figure 6*). Importantly, in both mRNA and tRNA export, Gle1/InsP$_6$ activation is required to stimulate ATPase activity and recycle the enzyme. In contrast, Dbp5 ATPase activity and Gle1 have been shown to be dispensable for pre-ribosomal subunit transport (*Neumann et al., 2016*). This raises interesting questions of whether RNA binding and nucleocytoplasmic shuttling of Dbp5 alone can achieve this function, where in the cell these interactions occur, and how rRNA substrates modulate Dbp5 activity.

In mRNA export, it has been proposed that Dbp5 displaces mRNA export factors such as Mex67 and Nab2 to promote directionality of mRNA transit (*Lund and Guthrie, 2005*; *Tran et al., 2007*). Observations that Mex67 and Dbp5 appear to both function parallel to Los1 to promote tRNA export may point to an overlapping pathway that shares both mRNA export factors. Dbp5 and Mex67 have been proposed to form physical interactions, with Mex67 serving as an adaptor to direct Dbp5 to appropriate mRNA sites for remodeling (*Lund and Guthrie, 2005*). However, here it is demonstrated that Mex67 is not required for Dbp5-tRNA interaction based on RNA IP assays. While Dbp5 recruitment to tRNA does not require Mex67, the data presented in this study do not exclude the possibility that the two proteins co-function to promote tRNA export. Indeed, previously published data suggests Mex67 likely requires an unknown adaptor to form its reported interactions with its tRNA substrates (*Yao et al., 2007*). Given the in vitro properties of Dbp5 binding to tRNA, it is tempting to speculate that nuclear Dbp5 promotes recruitment of Mex67 or Crm1 to tRNA-protein complexes. Indeed, a similar phenomenon has been described for the exon junction complex (EJC) and the DEAD-box protein at its core, eIF4AIII, that acts as a platform for the binding of other proteins to an mRNA (*Ballut et al., 2005*; *Andersen et al., 2006*; *Bono et al., 2006*; *Pacheco-Fiallos et al., 2023*). Stable binding of the EJC is governed by trans-acting proteins that lock the complex on the mRNA by stabilizing an ADP-P$_i$ hydrolysis immediate (*Ballut et al., 2005*; *Bono et al., 2006*; *Nielsen et al., 2009*; *Linder and Fuller-Pace, 2013*). In the case of Dbp5, the in vitro data suggests an inhibited state may be achieved in the context of tRNA alone, which could be further stabilized in vivo by yet to be discovered protein factors. The ability of Gle1/InsP$_6$ to induce maximal activity of Dbp5 in the presence of tRNA would then allow for resolution of stable Dbp5-tRNA complexes in a spatially regulated manner to terminate export and recycle bound export factors (*Figure 6*). Alternatively, given that it has been reported that Mex67 can bind 5′ extended tRNAs and that these transcripts have been previously reported to contain 5′ methyl-guanosine cap structures (*Ohira and Suzuki, 2016*; *Chatterjee et al., 2022*), perhaps an 'mRNA export-like' pathway that involves Cap Binding Complex (CBC) can act to recruit Mex67 to a subset of tRNAs. It remains unclear whether CBC interacts with such transcripts and whether other export factors like Dbp5 also support 'premature' export of unprocessed tRNAs.

More broadly, like many DEAD-box proteins, Dbp5 has many reported roles in controlling gene expression (*Estruch and Cole, 2003*; *Rollenhagen et al., 2004*; *Gross et al., 2007*; *Scarcelli et al., 2008*; *Tieg and Krebber, 2013*; *Wu et al., 2014*; *Neumann et al., 2016*; *Mikhailova et al., 2017*; *Lari et al., 2019*; *Beißel et al., 2020*). The nucleic acid substrate-specific biochemical properties of Dbp5 ATPase regulation defined here may have wide-ranging implications for Dbp5 functions in pre-ribosomal RNA export, translation, or R-loop metabolism. Furthermore, several DEAD-box proteins such as Dbp2 have been reported to bind diverse nucleic acid substrates (ncRNA and mRNA), including G-Quadraplexes, and have roles in R-loop biology (*Ma et al., 2016*; *Xing et al., 2017*; *Tedeschi et al., 2018*; *Gao et al., 2019*; *Lai et al., 2019*; *Lai and Tran, 2021*; *Yan et al., 2021*; *Song et al., 2022*). Given the high degree of conservation in the structure, function, and regulation of Dbp5 with other DEAD-box proteins (*Ozgur et al., 2015*; *Sloan and Bohnsack, 2018*), the observations reported here raise the possibility that other DEAD-box proteins may also demonstrate substrate-specific regulation. Furthermore, the results here support the importance of Gle1 and the small molecule InsP$_6$ as potent regulators of Dbp5 activity in multiple RNA export pathways. Thus, future investigation of how

Dbp5-mediated export is modulated by regulation of these cellular factors during stress or disease contexts represents important areas of future study.

Overall, this study and other recent publications support a general function for both Dbp5 and Mex67 in RNA export (*Yao et al., 2007*; *Yao et al., 2008*; *Faza et al., 2012*; *Wu et al., 2014*; *Neumann et al., 2016*; *Chatterjee et al., 2017*; *Becker et al., 2019*; *Lari et al., 2019*; *Nostramo and Hopper, 2020*; *Vasianovich et al., 2020*; *Chatterjee et al., 2022*). While these factors have been most well characterized in mRNA export, it is now important to reconsider Dbp5 and Mex67 as general RNA export factors, which raises questions about the possible co-regulation of mRNA and non-coding RNA export pathways to fine-tune gene expression. To this end, the in vitro, biochemical, and genetic properties of Dbp5 reported in this study will inform future structure–function and mechanistic characterization of a Dbp5-mediated, and Gle1/InsP$_6$-regulated, tRNA export pathway(s). For example, it will be important for future studies to address the composition of pathway-specific export-competent tRNA-protein complexes and how Dbp5 contributes to changes in the architecture of such a complex as they move from nucleus to cytoplasm and back again.

## Materials and methods

### Yeast strains generation and growth conditions

A list of all yeast strains used in this study is provided in *Supplementary file 1*. Deletion mutants and C-terminal tagging was conducted by PCR-based transformations and confirmed by colony PCR. All yeast transformations were conducted using previously published standard high-efficiency LiAc, ssDNA, PEG protocol (*Gietz and Woods, 2002*). Yeast were cultured in YPD or synthetic complete (SC) media when indicated to mid-log growth. For overexpression of Dbp5 ATPase mutants, untagged Dbp5 or *dbp5$^{R426Q}$*, *dbp5$^{R369G}$*, or *dbp5$^{E240Q}$* were integrated at URA3 locus under control of pGAL promoter in a wild-type strain with Dbp5 expressed from its endogenous locus and promoter. pGAL expression was induced by first derepressing the promoter with growth in YP media with 2% raffinose overnight followed by a shift to 2% galactose containing media for 6 hr. Overexpression was halted by addition of 2% glucose and further incubation for 1 hr.

### Spot growth assay

Yeast were pre-cultured to saturation overnight, diluted, and 1:10 serial dilutions were made with the most concentrated wells representing a concentration of 0.25 OD/ml. Then, 3 μl of each dilution was spotted on YPD and grown at indicated temperatures for 2 d.

### Live-cell imaging

Imaging experiments were carried out using the confocal configuration of an Andor Dragonfly microscope equipped with an EMCCD camera driven by Fusion software (Andor) using a ×60 oil immersion objective (Olympus, numerical aperture [N.A.] 1.4). Images were acquired from cells grown to mid-log phase in SC media at 25°C and shifted to 37°C for indicated time or treated with DMSO or 500 uM auxin and 10 uM InsP6 for relevant experiments. Cells were immobilized in 384-well glass-bottomed plates (VWR) that were pre-treated with concanavalin A.

### tRNA extraction and northern blotting

Isolation of small RNAs and northern blot experiments were performed as previously described (*Wu et al., 2013*). Briefly, small RNAs were isolated from yeast cultures grown to early log phase by addition of equal volumes cold TSE buffer (0.01 M Tris pH 7.5, 0.01 M EDTA, 0.1 M sodium chloride) and TSE saturated phenol. Samples were incubated at 55°C for 20 min with vortexing every 3 min, then placed on ice for 10 min. Aqueous phase was extracted after centrifugation, re-extraction with phenol was performed, and RNA was precipitated overnight in ethanol at –80°C. Then, 2.5 ug of RNA was separated on 10% TBE-Urea gels, gels were stained with Apex safe DNA gel stain to visualize 25S and 18S rRNA, and then transferred to Hybond N$^+$ membrane (Amersham). RNA was crosslinked to membranes at 2400 J/m$^2$ using UV crosslinker (VWR), and tRNA were detected using digoxigenin-labeled (DIG) probes. Mean integrated intensity values were measured for I and P bands using FIJI (*Schindelin et al., 2012*) and normalized to background signal in each lane.

The sequence of northern blot probe targeting precursor and mature isoforms of tRNA$^{\text{Ile}}_{\text{UAU}}$ used in *Figures 2 and 4* is as follows: GGCACAGAAACTTCGGAAACCGAATGTTGCTATAAGCACGAAGC TCTAACCACTGAGCTACACGAGC.

## Fluorescence in situ hybridization

FISH experiments were carried out as described in previous publications with minor modifications (*Lord et al., 2017*) using directly Cy3-labeled tRNA probes or fluorescein isothiocyanate (FITC)-labeled dT probes. The sequence of Cy3-labeled tRNA$^{\text{Ile}}_{\text{UAU}}$ intron-specific probe is same as previously published SRIM03 (*Chatterjee et al., 2017*) with Cy3 moiety appended at the 5′ end (CGTTGCTT TTAAAGGCCTGTTTGAAAGGTCTTTGGCACAGAAACTTCGGAAACCGAATGTTGCTAT). Briefly, yeast cultures were grown to early log phase at 25°C and treated with drug/vehicle or shifted to 37°C for 4 hr when appropriate. Samples were fixed with 0.1 volume of 37% formaldehyde for 15 min. These samples were then pelleted and resuspended in 4% paraformaldehyde (PFA) solution (4% PFA, 0.1 M potassium phosphate [pH 6.5], 0.5 mM MgCl$_2$) and incubated with rotation at 25°C for 3 hr. Cells were then washed twice with Buffer B (1.2 M sorbitol, 0.1 M potassium phosphate [pH 6.5], 0.5 mM MgCl$_2$). For spheroplasting, cell pellets were resuspended in a 1 ml Buffer B solution containing 0.05% β-mercaptoethanol. Then, 15 ul of 20 mg/ml zymolyase T20 was added to each sample and subsequently incubated at 37°C for 45 min. After washing once with Buffer B, cells were adhered to eight-well slides (Fisher Scientific) pre-coated with poly-L-lysine. Slides were placed sequentially in 70, 90, and 100% ethanol for 5 min and then allowed to dry. Samples were next incubated at 37°C for 2 hr in pre-hybridization buffer (4× SSC, 50% formamide, 10% dextran sulfate, 125 ug/ml *Escherichia coli* tRNA, 500 ug/ml salmon sperm DNA, 1× Denhardt's solution, and 10 mM vanadyl ribonucleoside complex [VRC]). Pre-hybridization solution was then aspirated and replaced with pre-warmed pre-hybridization solution that contained either 0.25 uM Cy3 tRNA probe or 0.025 uM FITC dT probe and further incubated at 37°C overnight. Wells were then sequentially washed for 5 min once with 2× SSC, three times with 1× SSC, and once with 0.5× SSC. Slides were dipped in 100% ethanol, air-dried, and mounting media with 4′,6-diamidino-2-phenyldole (DAPI) was applied to each well prior to sealing of coverslip onto slide. Imaging was performed using Andor Dragonfly equipped with Andor iXon Ultra 888 EMCCD camera driven by Fusion software (Andor) using a 60× 1.4 N.A. oil objective. Images were processed in FIJI (e.g., cropping, brightness/contrast adjustments, and maximum z-projections) (*Schindelin et al., 2012*). Quantification of tRNA FISH was performed by acquiring average nuclear and cytoplasmic pixel intensities, respectively. Maximum projection images were generated from z-stack images for DAPI and tRNA FISH channels. Nuclear and whole-cell masks were then respectively generated using Cellpose software (*Stringer et al., 2021*; *Pachitariu and Stringer, 2022*). Nuclear mask regions were removed from whole-cell masks to generate cytoplasmic mask and average pixel intensities were calculated. The ratio of average nuclear and cytoplasmic mask pixel intensity per cell was calculated.

## RNA immunoprecipitation

RIPs were conducted as previously described with minor modifications (*Lari et al., 2019*). Pull-downs were performed targeting protein-A (prA)-tagged Dbp5 in parallel with an untagged control to assess background non-specific binding. Crosslinking was conducted on yeast cultures in mid-log growth by addition of formaldehyde to final concentration of 0.3% and incubation for 30 min. Crosslinking was quenched by addition of glycine to a final concentration of 60 mM for 10 min. Cells were harvested and pellets frozen in liquid nitrogen. Lysis was performed using ice-cold TN150 (50 mM Tris–HCl pH 7.8, 150 mM NaCl, 0.1% IGEPAL, 5 mM β-mercaptoethanol) supplemented with 10 mM EDTA, 1 ng Luciferase Spike-In RNA (Promega), and 1× protease inhibitor cocktail. Then, 1 ml zirconia beads (0.5 mm) were used for disrupting cells by vortexing five times for 30 s followed by 1 min on ice between each pulse. Lysate was pre-cleared by centrifugation at 4000 × *g* 5 min followed by 20 min at 20,000 × g at 4°C. Also, 1% pre-IP lysate was preserved as input RNA sample and additional pre-IP sample was reserved for western blotting. Remaining lysates were then diluted to 10 ml with TN150 and incubated with IgG-conjugated magnetic dynabeads at 4°C for 30 min with constant rotation. Immunoprecipitate was washed once with 1 ml TN150, once with 1 ml TN1000 (50 mM Tris–HCl pH 7.8, 1 M NaCl 0.1% IGEPAL, 5 mM β-mercaptoethanol), and again once with 1 ml TN150 each for 5 min at 4°C. Beads from RIPs were then resuspended in 500 ul proteinase K elution mix (50 mM

Tris–HCl pH 7.8, 50 mM NaCl, 1 mM EDTA, 0.5% SDS, 100 ug proteinase K) in parallel with input samples and incubated at 50°C for 2 hr followed by 65°C for 1 hr to allow crosslink reversal. RNA was isolated by extraction with acidic phenol:chloroform:isoamylalcohol (pH 4.3–4.7) and ethanol precipitation. Half volume of RNA was reverse transcribed using Superscript III using random priming according to the manufacturer's instructions and other half of RNA was retained for minus RT control. RNA was analyzed by qPCR using Power SYBR (Applied Biosystems) on an Applied Biosystems instrument. Target RNA abundance in RIP was normalized to abundance of target in input sample and relative fold enrichment was calculated by comparing signal in specific prA-Dbp5 RIPs to signal from untagged control. Standard curves were generated to test PCR efficiencies of each primer set used in this study. Primer sequences used are as follows:

> $tRNA^{Ile}_{UAU}$ unspliced forward: GCTCGTGTAGCTCAGTGGTTAG
> $tRNA^{Ile}_{UAU}$ unspliced reverse: CTTTTAAAGGCCTGTTTGAAAG
> FBA1 forward: CGAAAACGCTGACAAGGAAG
> FBA1 reverse: TCTCAAAGCGATGTCACCAG

## Western blotting

Approximately 5 OD pellets were lysed in Laemmli buffer and loaded onto 10% SDS-PAGE. Protein transfer was performed onto nitrocellulose membrane using cold wet tank transfer protocol. Western blotting was performed with 1:5000 dilution of mouse monoclonal anti-DBP5 and 1:10,000 dilution of mouse monoclonal anti-GAPDH (Thermo Fisher) primary antibody overnight at 4°C and 1:10,000 anti-mouse DyeLight 650 secondary to detect protein.

## Protein purification, ATPase assays, and fluorescence polarization

Protein purification, in vitro ATPase assays, and fluorescence polarization using full-length Dbp5 were performed as described previously (*Weirich et al., 2006*; *Montpetit et al., 2011*; *Montpetit et al., 2012*). Commercially available poly (I:C), mixed yeast tRNA, and yeast phenylalanine tRNA were obtained from Sigma for use in indicated assays. For RNase T1 ATPase assays, indicated RNA samples were treated with 33 units of RNase T1 for 2 hr at 37°C prior to experiment. Fluorescence polarization experiments were performed using a fluorescein (fl)-labeled 16nt ssRNA (5′-fl-GGGUAAAAAAAAAAAA-3′) in the presence of ADP•BeF$_3$ with increasing concentrations of unlabeled polyadenylic acid or yeast mixed tRNA titrated into the reactions. Fluorescence polarization and ATPase assays were assembled in the same ATPase buffer (30 mM HEPES [pH 7.5], 100 mM NaCl, and 2 mM MgCl$_2$). All ATPase assays with RNA titrations were fit to Michaelis–Menten or allosteric sigmoidal models (when Gle1/InsP$_6$ was added) in GraphPad Prism.

## RNA EMSA and denaturing urea PAGE

Recombinant full-length Dbp5 was purified as above and described previously (*Montpetit et al., 2011*; *Montpetit et al., 2012*). RNA binding and EMSA were performed according to previously published assay conditions (*Yao et al., 2007*). RNA-binding reactions were assembled using binding buffer (20 mM HEPES [pH 7.4], 100 mM KCl, 10 mM NaCl, 4 mM MgCl$_2$, 0.2 mM EDTA, 20% glycerol, 1 mM DTT, 0.5% NP-40) in the presence of indicated nucleotide and loaded on to 6% native polyacrylamide gel (0.5× TBE). Gels were stained with Apex gel stain to visualize RNA. Mean integrated intensity of bands was quantified in FIJI (*Schindelin et al., 2012*) and normalized to background signal for each respective well. Bound fraction was calculated and fit to a one-site binding equation in GraphPad Prism.

## Acknowledgements

We acknowledge and thank all current and past members of the Montpetit lab for their aid and helpful discussions over the course of this work. In particular, we thank Rachel Montpetit for assistance with cloning and strain construction. We would also like to thank the members of the Aitchison and Wozniak laboratories for their constructive feedback during this study. AANR was funded in part by the predoctoral Training Program in Molecular and Cellular Biology at UC Davis that is supported by an NIH T32 training grant (GM007377). RA was supported by a postdoctoral fellowship from Uehara Memorial Foundation, and an overseas research fellowship from the Japan Society for the Promotion

of Science. Research was further supported by the National Institute of General Medical Sciences of the National Institutes of Health under Award Number R35GM145328. The content is solely the responsibility of the authors and does not necessarily represent the views of the funding agencies.

## Additional information

### Funding

| Funder | Grant reference number | Author |
|---|---|---|
| National Institute of General Medical Sciences | R35GM145328 | Ben Montpetit |
| National Institute of General Medical Sciences | GM007377 | Arvind Arul Nambi Rajan |
| Japan Society for the Promotion of Science | | Ryuta Asada |
| Uehara Memorial Foundation | | Ryuta Asada |

The funders had no role in study design, data collection and interpretation, or the decision to submit the work for publication.

### Author contributions

Arvind Arul Nambi Rajan, Conceptualization, Data curation, Formal analysis, Investigation, Methodology, Writing - original draft, Writing - review and editing; Ryuta Asada, Formal analysis, Methodology; Ben Montpetit, Conceptualization, Supervision, Funding acquisition, Investigation, Methodology, Project administration, Writing - review and editing

### Author ORCIDs

Arvind Arul Nambi Rajan http://orcid.org/0000-0003-3559-1421
Ryuta Asada http://orcid.org/0000-0001-8360-6355
Ben Montpetit http://orcid.org/0000-0002-8317-983X

Reviewer #1 (Public Review): https://doi.org/10.7554/eLife.89835.4.sa1
Reviewer #2 (Public Review): https://doi.org/10.7554/eLife.89835.4.sa2
Author Response https://doi.org/10.7554/eLife.89835.4.sa3

## Additional files

### Supplementary files

• Supplementary file 1. Supplementary table of yeast strains. List of yeast strains used in this study, along with associated genotypes and study of origin.
• MDAR checklist

### Data availability

Data generated or analysed during this study are available at https://github.com/montpetitlab/Rajan-et-al.-2023, (copy archived at *Montpetit Lab, 2023*).

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
