## [Editor Report · eLife assessment]

The work is a **valuable** contribution to understanding the mechanism of nuclear export of tRNA in budding yeast. The authors present **solid** evidence that Dbp5 functions in parallel with Los1 and Msn5 in tRNA export, in a manner dependent on Gle1 for activation of its ATPase activity but independently of Mex67, Dbp5's partner in mRNA export. It further presents biochemical evidence that Dbp5 can bind tRNA but that Gle1 and InsP6 are required for activating ATP hydrolysis by the Dbp5-tRNA complex, suggesting a possible mechanism for tRNA export by Dbp5.

---

## [Referee Report · Reviewer #1 (Public Review)]

This study focuses on the defining cellular pathways critical for tRNA export from the nucleus. While a number of these pathways have been identified, the observation that the primary transport receptors identified thus far (Los1 and Msn5) are not essential and that cells are viable even when both the genes are deleted supports the idea that there are as yet unidentified mediators of tRNA export from the nucleus. This study implicates the helicase Dbp5 in one of these parallel pathways arguing that Dbp5 works in a pathway that is independent of Los1 and/or Msn5. The authors present genetic data to support this conclusion. At least one results suggests that the idea of these pathways in parallel may be too simplistic as deletion of the LOS1 gene, which is not essential decreases the interaction of tRNA export substrate with Dbp5 (Figure 2A). If the two pathways were working in parallel, one might have expected removing one pathways to lead to an increase in the use of the other pathway and hence the interaction with a receptor in that pathway. The authors provide solid evidence that Dbp5 interacts with tRNA directly and that addition of the factor Gle1 together with the previously identified co-factor InsP6 can trigger helicase activity and release of tRNA. The combination of in vivo studies and biochemistry provide evidence to consider how Dbp5 contributes to export of tRNA and more broadly adds to the conversation about how coding and non-coding RNA export from the nucleus might be coordinated to control cell physiology.

---

## [Referee Report · Reviewer #2 (Public Review)]

In the manuscript by Rajan et al., the authors have highlighted the direct interaction between Dbp5 and tRNA, wherein Dbp5 serves as a mediator for tRNA export. This export process is subject to spatial regulation, as Dbp5 ATPase activation occurs specifically at nuclear pore complexes. Notably, this regulation is independent of the Los1-mediated pre-tRNA export route and instead relies on Gle1. The manuscript is well constructed and nicely written.

---

## [Author Response]

The following is the authors’ response to the previous reviews.

We would like to thank the Editors and Reviewers for their additional comments and constructive feedback on our manuscript. We have made minor adjustments to the figures and texts based on their suggestions, including improved images in Figure 1 and correction of figure labels.

**Reviewer #1 (Public Review):**
In their previous paper (Lari et al, 2019; Azra Lari Arvind Arul Nambi Rajan Rima Sandhu Taylor Reiter Rachel Montpetit Barry P Young Chris JR Loewen Ben Montpetit (2019) A nuclear role for the DEAD-box protein Dbp5 in tRNA export eLife 8:e48410.) as well as in the current manuscript the authors states that Dbp5 is involved in the export of tRNA that is independent of and parallel to Los1. They state that Dbp5 binds to the tRNA independent of known tRNA export proteins. The obtained conclusion is both intriguing and innovative, since it suggests that there are other variables, beyond the ones previously identified as tRNA factors, that might interact with Dbp5 to facilitate the export process. In order to find out additional factors aiding this process the authors may employ total RNA-associated protein purification (TRAPP) experiments ( Shchepachevto et al., 2019; Shchepachev V, Bresson S, Spanos C, Petfalski E, Fischer L, Rappsilber J, Tollervey D. Defining the RNA interactome by total RNA-associated protein purification. Mol Syst Biol. 2019 Apr 8;15(4):e8689. doi: 10.15252/msb.20188689. PMID: 30962360; PMCID: PMC6452921) to identify extra factors involved in conjunction with Dbp5. The process elucidates hitherto uninvestigated tRNA export components that function in conjunction with Dbp5.

Author Response: We greatly appreciate this suggestion and agree with the reviewer that identification of the composition of the export competent Dbp5 containing tRNA complex is a critical next step for understanding the mechanism of Dbp5 mediated tRNA export, which will form the foundation of a future investigation in the laboratory and warrants its own study.

**Reviewer #1 (Public Review):**
Various reports suggest that eukaryotic translation elongation factor 1 eEF1A is involved tRNA export Bohnsack et al., 2002 (Bohnsack MT, Regener K, Schwappach B, Saffrich R, Paraskeva E, Hartmann E, Görlich D. Exp5 exports eEF1A via tRNA from nuclei and synergizes with other transport pathways to confine translation to the cytoplasm. EMBO J. 2002 Nov 15;21(22):620515. doi: 10.1093/emboj/cdf613. PMID: 12426392; PMCID: PMC137205), (Grosshans etal., 2002; Grosshans H, Hurt E, Simos G). An aminoacylation-dependent nuclear tRNA export pathway in yeast.(Genes Dev. 2000 Apr 1;14(7):830-40. PMID: 10766739; PMCID: PMC316491). The presence of mutations in eEF1A has been seen to hinder the nuclear export process of all transfer RNAs (tRNAs). eEF1A has been shown to interact with Los1 aiding in tRNA export. The authors can also explore the crosstalk between Dbp5 and eEF1A in this study. Additionally, suppressor screening analysis in dbp5R423A , los1∆dbp5R423A los1∆msn∆dbp5R423A could shed more light on this.

Author Response: Thank you for this suggestion and raising an important possible role for Dbp5 in eEF1A mediated tRNA export. Based on more recent investigation of eEF1A function in tRNA export (PMID: 25838545), it is likely that eEF1A functions in re-export of charged tRNAs specifically (likely in conjunction with Msn5). The current manuscript has largely focused on the role of Dbp5 in pre-tRNA export, but a more careful mechanistic characterization of Dbp5 and re-export will be conducted in follow-up studies given the physical interaction between Dbp5 and spliced tRNAs we previously reported. Similarly, suppressor screens with the Dbp5 and los1Δmsn5Δ mutants will likely be a useful tool in identifying additional tRNA export factors and we thank the reviewer for this suggestion.

**Reviewer #1 (Public Review):**
The addition of Gle1 is potentially novel but it's unclear why the authors didn't address the potential involvement of IP6.

Author Response: The text has been revised to highlight the importance of InsP6 in Gle1 mediated activation of Dbp5. This includes referencing InsP6 throughout the manuscript during discussions of Gle1 activation of Dbp5 and lines 401-404 discussing the potential role for the small molecule in regulating mRNA and tRNA export in different cellular contexts (e.g., stress and disease).